# Citrate supplement facilitated muscle growth and renal maturation

**Michiaki Abe** [1,2]*, **Kazuhiko Kawaguchi**[3], **Satomi Yamasaki**[3], **Toshiki Nakai**[3], **Masaharu Hatachi**[3], **Takanori Mizuno**[3], **Atsuko Masaura**[1], **Naonori Kumagai**[4], **Tadashi Ishii**[1]

1 Department of Education and Support for Regional Medicine, Tohoku University Hospital, Sendai, Miyagi, Japan, 2 Tohoku Medical Megabank Organization, Tohoku University, Sendai, Miyagi, Japan, 3 Medical Affairs Department, Nippon Chemiphar Co., Ltd., Chiyoda-ku, Tokyo, Japan, 4 Department of Pediatrics, Fujita Health University, Toyoake, Aichi, Japan

* michiaki.abe.a1@tohoku.ac.jp

## Abstract

Citrate supplementation is well known to alleviate muscle fatigue. Furthermore, our previous clinical study revealed that citrate supplementation prevents renal oxidative stress and dysfunction. We hypothesize that an interaction between muscle and kidney tissues underlies the effects of citrate supplementation. This study investigated the effects of a citrate agent, potassium citrate/sodium citrate (PCSC), on muscle and renal functions. Male Sprague-Dawley rats were randomly assigned to two groups (control and PCSC groups). PCSC (2000 mg/kg body weight) was administered orally administrated for one week. Kidney weight, vastus lateralis muscle weight, and renal function were compared between the groups. Subsequently, the kidney and muscle tissues were analyzed using metabolomics. The PCSC group showed a significant increase in muscle mass relative to the weight gain ($p = 0.0472$). Renal function development with growth was more pronounced in the PCSC group ($p < 0.0001$). Metabolomic analysis of muscle tissue in the PCSC group revealed increased alanine levels and decreased levels of sarcosine, creatinine, and NADPH/NADP+ ratio. In the kidney tissue, PCSC supplementation led to elevated N,N-dimethylglycine, urea, and the ratio of malic acid to aspartate, while betaine aldehyde, carnitine, and Fisher's ratio decreased. The study concluded that PCSC supplement facilitated muscle growth metabolically through an alanine-associated pathway and renal function development by increasing intrarenal urea and accelerating the malate-shuttle and the betaine pathway. These findings indicate PCSC's potential impact of PCSCs on muscle-kidney interactions.

**Data availability statement:** All relevant data are within the paper and its Supporting Information files.

**Funding:** This study was supported by "Collaboration research fund with Nippon Chemiphar Co., Ltd. (Tokyo, Japan, https://www.chemiphar.co.jp/english/)" and "JSPS KAKENHI Grant (C) 22K11849 from Japan Society for the Promotion of Science (https://www.jsps.go.jp/j-grantsinaid/)". Material support: Potassium citrate/sodium citrate (PCSC) was kindly provided by Nippon Chemiphar Co., Ltd. (Tokyo, Japan, https://www.chemiphar.co.jp/english/). Some employees of the funding company contributed to study design, data collection and data analysis.

**Competing interests:** Michiaki Abe collaborated with Nippon Chemiphar Co. Ltd. (Tokyo, Japan). Kazuhiko Kawaguchi, Satomi Yamasaki, Toshiki Nakai, Takanori Mizuno and Masanori Hatachi are all employed by the company.

## Introduction

Citrate supplementation is considered as improving exercise performance [1,2]. Exercise generates waste products in the body, which are eliminated through urine. Renal dysfunction leads to the accumulation of uremic toxins and metabolic acidosis. Metabolic acidosis is a risk factor for the progression of cardiovascular diseases and a contributing factor in the vicious cycle of chronic kidney disease (CKD) [3]. Renal dysfunction causes fatigue and muscle weakness. Recently, we demonstrated that chronic administration of potassium citrate/sodium citrate (PCSC) reduces intrarenal oxidative stress in patients with mild CKD [4]. We hypothesized that PCSC supplement to patients with normal renal function would prevent the onset of CKD.

On the other hand, sarcopenia is prevalent in CKD patients and is considered to be related to decreased mitochondrial function [5]. In this study, we investigated whether PCSC could support muscle and renal functions in normal rodents. Furthermore, the muscular-renal interaction was evaluated by metabolomic analysis of muscle and kidney tissues after PCSC administration.

## Materials and methods

### Animals and experimental protocol

Male Crl:CD(SD) rats (age, 8 weeks; body weight, 255–285 g) obtained from Charles River Laboratories Japan, Inc. (Tokyo, Japan) were randomly divided into two groups (n = 8 each): PCSC and control. The PCSC group received 2000 mg/kg body weight of PCSC by oral gavage twice a day for one week. The dose was decided according to the previous reports [6,7], whereas the control group received water. Venous blood and 21-hour urine were collected on day 0 and day 8. On day 8 following the experimental protocol, sacrifice was performed by euthanasia under isoflurane anesthesia (Mylan Japan Inc., Tokyo, Japan). The weights of the vastus lateralis (VL) and kidneys were measured bilaterally. All research staff involved in animal care received specialized training in accordance with the ARRIVE guideline and all animals were welfare throughout the protocol.

### Histological examination

After the sacrifice, the right kidney and the right VL muscle from the control and PCSC groups were fixed in 10% neutral buffered formalin. The tissues were then paraffin-embedded, sectioned, and stained with hematoxylin and eosin for microscopic examination.

### Extraction of metabolites and preparations for metabolome analyses

The left kidney and left VL muscle from each rat in both groups were stored at −80°C. Frozen tissues were homogenized in methanol containing internal standards (20 µmol/L methionine sulfone, 20 µmol/L 2-(N-morpholino) ethane sulfonic acid, and 40 µmol/L D-camphor-10-sulfonic acid). After centrifugation at 10,000 × $g$ for 10 min, the supernatant was mixed with equal volumes of chloroform and 1/2.5 volume of deionized water. Insoluble materials were removed by centrifugation at 10,000 × $g$

for 10 min, and the upper aqueous layer was filtered using an Amicon Ultrafree-MC ultrafilter (cut-off = 3,000 Da. Milli-pore Co., Billerica, MA) The filtrates were dried and stored at −80°C until assay. Before analysis, each metabolic extract was dissolved in deionized water containing reference compounds (200 µmol/L each of 3-aminopyrrolidine and trimesic acid) to correct for migration times. Metabolome analyses were conducted using Human Metabolome Technologies Inc. (Tsuruoka, Japan). Capillary electrophoresis time-of-flight mass spectrometry (CE–TOFMS) was performed as described previously [8].

## Statistics

Kidney weights and VL muscle weights (MW) are calculated by the formula: Organ weight (g/gBW) = [weight of organ]/[body weight on day 8]. Renal function was calculated and presented as 24-hour creatinine clearance (24hrCcr, ml/min). Statistical analysis was performed using JMP Pro17 (JMP Statistical Discovery LLC.). Significant differences of body weight, food consumption, kidney weight, VL muscle weight and renal function were defined as $p < 0.05$ by Student's *t*-test. Significant differences of metabolomics were defined as $p < 0.05$ by Welch's *t*-test.

## Ethics statement

Animal use protocols were approved by the Animal Care Committee of Kagohbutsu-Anzensei Institute (Sapporo, Japan) according to national guidelines (Regulations and Guidelines on Scientific Care and Use of Laboratory Animals in Japan), which were recognized by the ARRIVE guidelines and were conducted within the institute. Animal IRB: AN20190618−04.

## Results and discussion

### Effects of PCSC on muscle mass and renal function

Body weight gain and food consumption were similar in the PCSC and control groups. There were no significant differences in the kidney weights between the groups. However, the MW at day 8 was significantly higher in the PCSC group than in the control (mean±SD: 0.096±0.013 g/BWg vs. 0.107±0.020 g/BWg, p=0.0449) (Fig 1 and Table 1). This suggested that PCSC administration facilitated muscle gain during the experimental period.

Renal function (24hrCcr) at day 0 (2.401 ml/min±0.185) was increased to that at day 8 (3.030±0.601) in the control group ($p=0.0027$), but a more pronounced increment was observed in the PCSC group from 2.391 ml/min±0.275 to

**(A) MW per body weight (g/100gBW)**   **(B) Kidney per body weight (g/100gBW)**

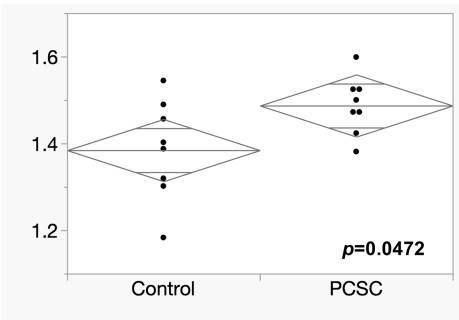 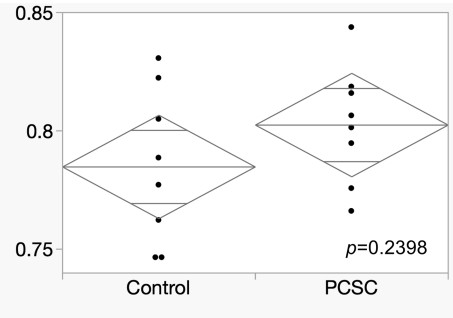

**Fig. 1. Comparison of muscle weight, muscle growth (MG) and kidney weight.** Comparison of the weights of bilateral kidneys and bilateral vastus lateral muscles between the Control and PCSC groups **(A and B)**. Comparison of muscle growth during the protocol period between the groups **(C)**. Significance was defined as $p < 0.005$ by *t*-test. The mean diamond indicates mean and 95% confidence intervalls.

**Table 1. Comparison of Body weight, Food consumption, Weights of Kidney and Vastus lateralis muscle and Renal function between Control and PCSC.**

| Body weight and Food consumption | | Body weight, g | | | | | Food consumption, g/rat/day | | |
|---|---|---|---|---|---|---|---|---|---|
| | | Day 1 | Day 3 | Day 5 | Day 7 | Day 8 | Day 2 | Day 4 | Day 7 |
| Control group | n | 8 | 8 | 8 | 8 | 8 | 8 | 8 | 8 |
| | mean | 272.1 | 288.6 | 302.8 | 316.3 | 319.0 | 25.94 | 27.60 | 27.31 |
| | SD | 6.38 | 8.47 | 8.24 | 9.97 | 9.91 | 2.085 | 2.282 | 1.755 |
| PCSC group | n | 8 | 8 | 8 | 8 | 8 | 8 | 8 | 8 |
| | mean | 269.9 | 286 | 302 | 317.1 | 314.9 | 25.34 | 26.41 | 27.34 |
| | SD | 9.14 | 9.41 | 9.13 | 10.22 | 9.63 | 1.790 | 1.811 | 1.538 |
| Control vs. PCSC | $p$ | 0.5771 | 0.5669 | 0.8009 | 0.9033 | 0.3739 | 0.5467 | 0.2682 | 1.0000 |
| Kidney and Muscle | | Renal function, mL/min | | Kidneys weight, g | Kidney weight per BW, | VL muscles weight, g | Muscle weight per BW, | | |
| | | 24hr.Ccr Day1 | 24hr.Ccr Day8 | Day 8 | g/100gBW | Day 8 | g/100gBW | | |
| Control group | n | 8 | 8 | 8 | 8 | 8 | 8 | | |
| | mean | 2.40 | 3.03 | 2.50 | 0.008 | 4.43 | 1.38 | | |
| | SD | 0.185 | 0.601 | 0.106 | 0.106 | 0.383 | 0.115 | | |
| PCSC group | n | 8 | 8 | 8 | 8 | 8 | 8 | | |
| | mean | 2.39 | 3.21 | 2.53 | 0.008 | 4.68 | 1.49 | | |
| | SD | 0.275 | 0.184 | 0.093 | 0.093 | 0.157 | 0.067 | | |
| Control vs. PCSC | $p$ | 0.9364 | 0.4450 | 0.6397 | 0.2398 | 0. 1133 | *0.0472** | | |

The values are mean and standard deviation of each variable.

PCSC, potassium citrate/sodium citrate; VL, vastus lateralis; 24hr.Ccr, 24-hour Creatinine Clearance.

* Significantly value, $p < 0.05$ by Student's *t-test*

$3.204 \pm 0.182$ ($p < 0.0001$), as shown in Fig 2. This suggests that PCSC facilitated renal function more stable than the controls.

We proposed that PCSC promoted exercise performance and muscle growth, which were suppressed by maturing renal function. This enhanced renal capacity leads to more efficient excretion of metabolic waste products and uremic toxins, thereby alleviating exercise-induced fatigue and creating a favorable environment for muscle anabolism. Our metabolomic analysis provides several insights into the underlying mechanisms.

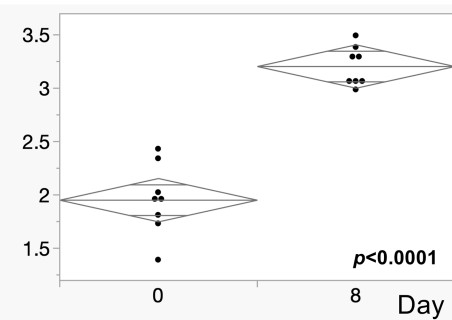

**Fig. 2. Comparison of 24-hour creatinine clearance (24hrCcr) of control and PCSC groups between day 0 and day 8.** Significance was defined as $p < 0.005$ by *t-test*. The mean diamond indicates mean and 95% confidence intervalls.

## Histological analysis of kidney and muscle

No abnormal histological findings were observed in the VL muscle or kidneys of either group (Fig 3). It was considered that 2000 mg/kg body weight of PCSCs was safe for muscle and kidney.

## Metabolome analysis of the muscle affected by PCSC

Metabolomic analysis of the VL muscles was performed to elucidate the mechanisms by which PCSC facilitate muscle growth (Table 1). PCSC significantly decreased sarcosine from 77.1 nmol/g ± 15.0 to 63.6 ± 6.4 ($p = 0.0353$) and creatinine from 94.7 nmol/g ± 7.0 to 87.3 ± 6.3 ($p = 0.0425$). Alanine levels significantly increased from 3221 ± 360 nmol/g to 3868 ± 730 nmol/g ($p = 0.0212$). In muscle metabolic pathways, the NADPH/NADP$^+$ ratio was significantly decreased from 1.49 ± 0.42 to 1.09 ± 0.18 ($p = 0.0263$).

   PCSC administration led to an increase in alanine levels in the muscle. This likely results from increased protein catabolism due to heightened physical activity and enhanced glycolysis. Given that muscles lack glucose-6-phosphatase, elevated alanine levels are transported to the liver. It activates the glucose-alanine cycle and stimulates gluconeogenesis, consequently increasing the glucose supply to the muscles. This amplified energy availability likely supports the observation of enhanced performance. Furthermore, alanine can be converted to glutamate and aspartate, which are utilized in the urea cycle, thereby increasing urea production. The observed increase in citrate activates the TCA cycle, whereas increased citrulline further promotes the hepatic urea cycle. [9] Increased urea is transported to the kidneys where it plays a crucial role in renal medullary osmolarity. Additionally, elevated alanine can be converted to pyruvate via gluconeogenesis, potentially activating the TCA cycle within the muscle itself.

## Metabolome analysis of the kidney affected by PCSC

Metabolomic analysis of the kidneys was performed to elucidate the mechanisms by which PCSC facilitate renal function development (Table 1). PCSC increased N,N-Dimethylglycine from 30.3 ± 6.6 nmol/g to 40.6 ± 10.3 ($p = 0.0322$) and malic acid from 347.2 ± 108.5 nmol/g to 497.8 ± 80.7 ($p = 0.0071$), and deceased betaine aldehyde from 25.0 ± 3.5 nmol/g

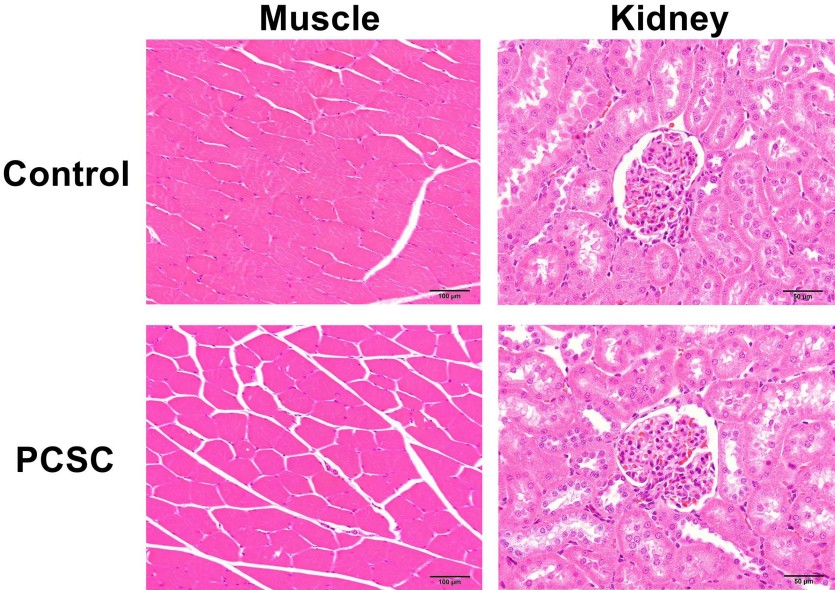

**Fig 3. Histological analysis.** Muscle and kidney of control and PCSC were evaluated by Hematoxylin-Eosin Stain. The indicator bar is 100μM.

to 17.0±3.4 ($p$=0.0004) and carnitine from 682.5±93.4 nmol/g to 581.5±73.9 ($p$=0.0309). Urea was increased from 16871.8±2900.0 nmol/g to 22801.9±5284.8 ($p$=0.0147). In kidney metabolic pathways, Fisher's ratio was decreased by PCSC from 2.77±0.07 to 2.61±0.18 ($p$=0.0292), and malic acid/aspartate (Mal/Asp) ratio was increased from 0.21±0.07 to 0.30±0.04 ($p$=0.0088).

Significant increases in the malate level and the Mal/Asp ratio were observed in the kidneys. In the kidney, cytoplasmic citrate can also be converted to malate by these enzymes. The increased malate/aspartate ratio suggests that PCSC could activate the malate-aspartate shuttle in renal mitochondria. This mechanism aligns with findings in a renal ischemia-reperfusion model where the inhibition of malate dehydrogenase-2 attenuated oxidative stress and protected tubular cells by shifting malate metabolism from the mitochondria to the cytosol via the malate shuttle [9]. Furthermore, a study in normal rats on high salt intake suggested that the malate-aspartate shuttle enhances energy supply by efficiently delivering cytosolic malate into the mitochondrial matrix, a change associated with suppressed TCA cycle and increased glycolytic enzyme transcription [10]. The malate-aspartate shuttle is considered to more contributes to mitochondrial ATP production in the kidney.

Next, we considered betaine metabolism, which is pertinent because betaine is abundant in the kidneys and aids in resisting hyperosmotic stress. In this study, betaine aldehyde was decreased, and dimethylglycine (DMG) was increased, suggesting the activation of betaine-homocysteine methyltransferase (BHMT). Although BHMT activity is highest in the liver, it is also expressed in the kidney. [10] The betaine-homocysteine metabolic pathway contributes to intrarenal osmolarity as well as the urea concentration. The antioxidant effect of DMG is also a potential benefit to the kidney.

**Summary of metabolomic analysis with significant pathways (Table 2, S1 and S2 Tables)**

**A: High level of significance ($p$<0.01)** A marked increase in malate (elevated Mal/Asp ratio) and decrease in betaine aldehyde were observed in the kidneys.
**B. The next higher significance (0.01<$p$<0.03):** urea was increased, and the Fischer ratio decreased in the kidneys. Additionally, alanine increased and the NADPH/NADP+ ratio decreased in the muscles.
**C. Low level of significance (0.03<$p$<0.05):** N,N-dimethylglycine was increased, and carnitine decreased in the kidneys. Sarcosine and creatinine levels decreased in the muscles.

Collectively, these data suggest a beneficial muscular-renal interaction, likely involving the liver. This interaction, potentiated by PCSC, appears to suppress the accumulation of uremic toxins in muscle via rapid processing of renal excretion, resulting in muscle strengthening. Previous studies have shown that high-intensity interval exercise can restore muscle mitochondrial function in rats with CKD [11]. Some uremic toxins impair mitochondrial health in CKD mice [12]. CKD-induced sarcopenia is caused by skeletal muscle atrophy [13]. Our findings suggest that PCSC could similarly facilitate recovery from exercise-induced muscle fatigue and promote muscle anabolism by maturing renal function. Further investigation is warranted to determine whether PCSC can prevent sarcopenia in elderly people.

The reduction of the muscle NADPH/NADP+ ratio, sarcosine, and creatinine levels by PCSC suggests increased oxidative stress due to exercise promotion. This finding was not inconsistent with the observation that bata-agonists increase sarcosine and creatinine, thereby preventing muscle atrophy in diabetic animal models [14]. The PCSC induced enhancement of muscle metabolism, simultaneously it increased renal urea concentration and activated the betaine-homocysteine metabolic pathway in the kidneys and possibly the liver. Thereby renal function could be improved, and the elimination of the toxins could be promoted. Thus, PCSC is hypothesized to contribute to the improvement of renal function through a collaboration between the muscle and the liver.

This study had some limitations. The weight of the VL muscles at baseline was unknown. To fully understand the muscle-kidney metabolic interaction, we must elucidate the role of the liver in this axis.

**Table 2. Significant Metabolites of Vastus lateralis muscle or Kidney. Control vs. PCSC.**

| Compound name, nmol/g | Muscle | | | | | | | Kidney | | | | | | |
|---|---|---|---|---|---|---|---|---|---|---|---|---|---|---|
| | Control | | | PCSC | | | | Control | | | PCSC | | | |
| | n | mean | SD | n | mean | SD | p | n | mean | SD | n | mean | SD | p |
| NAD+ | 8 | 378.2 | 42.77 | 8 | 425.3 | 56.00 | 0.0791 | 8 | 526.2 | 50.10 | 8 | 491.2 | 29.90 | 0.1119 |
| NADH | 8 | 35.1 | 5.28 | 8 | 37.8 | 5.34 | 0.3176 | 8 | 22.6 | 2.05 | 8 | 20.9 | 1.71 | 0.0957 |
| NADP+ | 8 | 11.2 | 2.13 | 8 | 13.8 | 2.65 | 0.0445 | 8 | 67.1 | 9.91 | 8 | 60.2 | 6.77 | 0.1283 |
| NADPH | 8 | 16.0 | 2.07 | 8 | 14.7 | 1.64 | 0.1970 | 8 | 17.5 | 2.22 | 8 | 17.1 | 2.21 | 0.7359 |
| **Malic acid** | 8 | 624.3 | 206.12 | 8 | 639.5 | 245.98 | 0.8952 | 8 | 347.2 | 108.50 | 8 | 497.8 | 80.70 | **0.0071\*** |
| **Urea** | 8 | 5725.8 | 632.51 | 8 | 6135.5 | 1060.25 | 0.3639 | 8 | 16871.8 | 2900.04 | 8 | 22801.9 | 5284.83 | **0.0147\*** |
| **Sarcosine** | 8 | 77.07 | 14.989 | 8 | 63.60 | 6.412 | **0.0353\*\*** | 8 | 30.70 | 4.960 | 8 | 33.90 | 10.070 | 0.4264 |
| **Ala** | 8 | 3120.7 | 359.71 | 8 | 3867.5 | 730.02 | **0.0212\*** | 8 | 1691 | 205.26 | 8 | 1826.1 | 173.83 | 0.1775 |
| ***N,N*-Dimethylglycine** | 8 | 10.4 | 1.87 | 8 | 12.6 | 2.99 | 0.0968 | 8 | 30.3 | 6.55 | 8 | 40.6 | 10.30 | **0.0322\*** |
| **Creatinine** | 8 | 94.7 | 6.96 | 8 | 87.3 | 6.33 | **0.0425\*\*** | 8 | 81.5 | 11.04 | 8 | 83.7 | 16.12 | 0.7523 |
| **Betaine aldehyde** | | | | | | | | 8 | 25.0 | 3.46 | 8 | 17.0 | 3.42 | **0.0004\*\*** |
| **Asp** | 8 | 147.1 | 69.07 | 8 | 178.5 | 65.04 | 0.3651 | 8 | 1698.7 | 244.49 | 8 | 1673.7 | 177.79 | 0.8188 |
| **Carnitine** | 8 | 787.8 | 125.70 | 8 | 867.7 | 154.54 | 0.2756 | 8 | 682.5 | 93.37 | 8 | 581.5 | 73.91 | **0.0309\*\*** |
| **NADPH/NADP+** | 8 | 1.49 | 0.418 | 8 | 1.09 | 0.179 | **0.0263\*\*** | 8 | 0.27 | 0.051 | 8 | 0.29 | 0.047 | 0.3882 |
| NADH/NAD+ | 8 | 0.09 | 0.018 | 8 | 0.09 | 0.018 | 0.7163 | 8 | 0.04 | 0.005 | 8 | 0.04 | 0.003 | 0.7797 |
| **Fischer's Ratio** | 8 | 1.95 | 0.236 | 8 | 1.68 | 0.300 | 0.0648 | 8 | 2.77 | 0.073 | 8 | 2.61 | 0.180 | **0.0292\*\*** |
| **Malate/Asp** | 8 | 4.84 | 1.877 | 8 | 3.96 | 2.070 | 0.3908 | 8 | 0.21 | 0.072 | 8 | 0.20 | 0.043 | **0.0088\*** |

The values are mean and standard deviation of each variable.

PCSC, potassium citrate/sodium citrate; NAD+, oxidized Nicotinamide Adenine Dinucleotide; NADH, reduced Nicotinamide Adenine Dinucleotide; NADP+, Nicotinamide Adenine Dinucleotide Phosphate; NADPH, reduced Nicotinamide Adenine Dinucleotide Phosphate; Ala, Alanine; Asp, Aspartic Acid.

\* Significantly increased with PCSC, $p < 0.05$ by Welch's *t*-test

\*\*Significantly decreased by PCSC, $p < 0.05$ by Welch's *t*-test

## Conclusion

This study demonstrated that PCSC supplement significantly facilitates both muscle mass and renal function maturation during body development. PCSC was conceded to improve renal function due to intrarenal osmolar concentration and activate the malate-aspartate shuttle and the betaine-homocysteine metabolic pathway. The improvement promotes the excretion of uremic toxins, that might reduce muscle fatigue and increase muscle mass. PCSC is thought to metabolically facilitate muscle and renal function.

## Supporting information

**S1 Table. Metabolites of Vastus Lateralis Muscle and Kidney.** Comparison of Control and PCSC. \* Significantly increased with PCSC, $p < 0.05$ by Welch's t-test. \*\*Significantly decreased by PCSC, $p < 0.05$ by Welch's *t*-test. (XLSX)

**S2 Table. Comparison of Metabolites of of Vastus Lateralis Muscle and Kidney.** Related with Methionine Cycle. \* Significant differences on comparison of kidney and muscle, $p < 0.05$ by Welch's *t*-test. (XLSX)

**S3 Table. 116 Metabolites in the kidney.** *p*-value by Welch's *t*-test (\* < 0.05, \*\* < 0.01, \*\*\* < 0.001). (XLSX)

**S4 Table. 116 Metabolites in the vastus lateralis muscle.** *p*-value by Welch's *t*-test (*<0.05, **<0.01, ***<0.001).
(XLSX)

**S5 Table. Source of the data of body weight, food consumption, kidney weight, vastus lateralis muscle weight and renal function (Ccr).**
(XLSX)

## Acknowledgments

The authors would like to acknowledge all players involved in this study for their participation. We express our gratitude to Kagohbutsu-Anzensei Institute for Animal Protocols and Human Metabolome Technologies Inc. for the metabolome analysis. We thank Mr. Dennis Nishimura for the English proofreading.

## Author contributions

**Conceptualization:** Michiaki Abe, Naonori Kumagai.

**Data curation:** Michiaki Abe, Satomi Yamasaki, Atsuko Masaura.

**Formal analysis:** Michiaki Abe, Masaharu Hatachi, Takanori Mizuno.

**Funding acquisition:** Michiaki Abe, Toshiki Nakai.

**Investigation:** Kazuhiko Kawaguchi, Satomi Yamasaki.

**Methodology:** Michiaki Abe, Kazuhiko Kawaguchi.

**Project administration:** Michiaki Abe, Tadashi Ishii.

**Resources:** Masaharu Hatachi, Atsuko Masaura.

**Supervision:** Toshiki Nakai, Tadashi Ishii.

**Writing – original draft:** Michiaki Abe.

**Writing – review & editing:** Naonori Kumagai.

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
