## [Decision Letter · Decision Letter 0]

14 Aug 2025

Dear Dr. Abe,

Thank you for submitting your manuscript to PLOS ONE. After careful consideration, we feel that it has merit but does not fully meet PLOS ONE’s publication criteria as it currently stands. Therefore, we invite you to submit a revised version of the manuscript that addresses the points raised during the review process.

We look forward to receiving your revised manuscript.

Kind regards,

Sudarshan Kasireddy, Ph.D.

Academic Editor

PLOS ONE

Journal Requirements:

2. To comply with PLOS One submissions requirements, in your Methods section, please provide additional information regarding the experiments involving animals and ensure you have included details on (1) methods of sacrifice, and (2) efforts to alleviate suffering.

3. Thank you for stating the following in the Competing Interests and Financial Disclosure section: [Michiaki Abe collaborated with Nippon Chemiphar Co. Ltd. (Tokyo, Japan).Kazuhiko Kawaguchi, Satomi Yamasaki, Toshiki Nakai, Takanori Mizuno and Masanori Hatachi are all employed by the company.].

We note that you received funding from a commercial source: [Nippon Chemiphar Co. Ltd]

Within this Competing Interests Statement, please confirm that this does not alter your adherence to all PLOS ONE policies on sharing data and materials by including the following statement: ""This does not alter our adherence to PLOS ONE policies on sharing data and materials.” (as detailed online in our guide for authors http://journals.plos.org/plosone/s/competing-interests).  If there are restrictions on sharing of data and/or materials, please state these. Please note that we cannot proceed with consideration of your article until this information has been declared.

4.  Thank you for stating the following financial disclosure:[ Collaboration research fund with Nippon Chemiphar Co., Ltd. (Tokyo, Japan). JSPS KAKENHI Grant (C) 22K11849.].

6. Please include captions for your Supporting Information files at the end of your manuscript, and update any in-text citations to match accordingly. Please see our Supporting Information guidelines for more information: http://journals.plos.org/plosone/s/supporting-information .

Reviewers' comments:

Reviewer's Responses to Questions

**Comments to the Author**

1. Is the manuscript technically sound, and do the data support the conclusions?

Reviewer #1: Yes

Reviewer #2: Partly

Reviewer #3: Yes

2. Has the statistical analysis been performed appropriately and rigorously?

Reviewer #1: Yes

Reviewer #2: Yes

Reviewer #3: Yes

3. Have the authors made all data underlying the findings in their manuscript fully available?

Reviewer #1: Yes

Reviewer #2: Yes

Reviewer #3: Yes

4. Is the manuscript presented in an intelligible fashion and written in standard English?

Reviewer #1: No

Reviewer #2: Yes

Reviewer #3: Yes

Reviewer #1: For reproducibility, clarifications are needed for the queries

Authors report on the role of potassium/sodium citrate in muscle growth and renal maturation

Abstract

Introduction under abstract lacks clarity and sentences are disjointed

Method

Authors administered 2000 mg/kg body weight of PCSC by oral gavage to PCSC group daily for one week. What is the rationale for choosing 2000mg/kg/bw?

Why did the authors choose 8 days for the drug/agent administration?

Any reason why authors used only one dose level rather than 3-5 to examine dose-dependency as a proof of cause and effect?

Statistics

Authors calculated muscle growth with the formula: MG= (weight of 70 vastus lateralis muscle)/[(body weight on day 8)- (body weight on day 0)], kindly provide reference

Formula for determining the renal function should be provided

Results and discussion

Kindly provide brief explanation for box plot. Is the middle line mean or median, the error bars, what quartile if it applies?

Reference appropriate sections eg., A significant increase in malate levels was observed in the kidneys. Cytoplasmic citrate can be 136 converted to malate by malate enzymes 1 and 3, utilizing NADH as a cofactor. The increased 137 malate/aspartate ratio suggests activation of the malate-aspartate shuttle in renal mitochondria.

Reviewer #2: Specific comments:

Abstract

• Please ensure that all abbreviations are spelled out in full the first time they appear in the abstract for clarity. This helps readers unfamiliar with the terms to better understand your work

• The abstract/background section appears limited in scope and depth. I recommend expanding it to provide a clearer context for the study, including key motivations, and the specific research gap being addressed. This will help readers better understand the significance and novelty of your work from the outset.

Introduction

• The introduction is currently too brief and does not sufficiently establish the physiological or mechanistic link between renal dysfunction and muscle fatigue. To improve the manuscript, I recommend expanding the introduction to include:

• (1) the role of kidney function in metabolic homeostasis,

• (2) how impaired renal function can lead to systemic metabolic disturbances

• (3) the downstream effects of these disturbances on skeletal muscle metabolism and fatigue.

(decreased creatinine and NADPH/NADP⁺ ratio may reflect altered energy metabolism and oxidative stress in muscle tissue, while changes in urea and amino acid metabolism (e.g., N,N-dimethylglycine, sarcosine) in kidney tissue could signal impaired nitrogen handling and mitochondrial dysfunction, both of which are known to contribute to fatigue)

material and methods

Animals and experimental protocol

• reference for 2000 mg/kg body weight of PCSC

Extraction of metabolites and preparations for metabolome analyses

• The section on “Extraction of metabolites and preparations for metabolome analyses” requires additional methodological detail to ensure reproducibility and transparency. The Method section should be expanded to include detailed information on animal handling and authentication.

Results

• I noticed that the pathology figure, which provides valuable visual evidence supporting the study’s findings, is currently included only in the supplementary materials. I strongly recommend moving this figure into the main manuscript. Specifically, it would be appropriate to reference and describe it within both the Methodology and Results sections as separate paragraphs.

Discussion

• Please enrich the Discussion section by adding references to previous studies on octreotide and deferoxamine to better support and contextualize your findings. Including relevant literature will help demonstrate how your results align with or differ from prior work, highlighting the novelty and significance of your study.

Reviewer #3: The paper proposes an experimental study with an easy design, yet results are presented in a convincing manner, I specifically appreciate that some biochemical parameters there are not usually assed have been evaluated by the current study (such as sarcosine or NADPH/NADP+ ratio). The supplementary files, especially the one in Excel-format, suggest a consistent, thoroughly done work.

Into the discussion section, the results are related to other studies in the medical literature, even though there are not many other on this topic. I appreciate the concentration on the mechanisms of kidney protection induced by citrate supplementation.

The section referring to limitations clearly and honestly states some of flaws of the model that researchers used.

**Do you want your identity to be public for this peer review?** For information about this choice, including consent withdrawal, please see our Privacy Policy

Reviewer #1: No

Reviewer #2: No

Reviewer #3: **Yes:** Diana Ciubotariu

---

## [Author Response · Author response to Decision Letter 1]

20 Nov 2025

Response to Reviewers

PONE-D-25-36940

Citrate supplement facilitated muscle growth and renal maturation.

We thank the reviewers for their thoughtful and constructive comments, which have significantly helped to improve the clarity and rigor of our manuscript. Our point-by-point responses to the reviewers' comments are provided below. All changes in the manuscript, including the main text, Tables, and Supplementary Tables, are highlighted in yellow.

We confirm that there are no issues regarding dual publication, research ethics, or publication ethics.

Responses to the specific comments:

All changed parts in the manuscript are highlighted in yellow.

1. PLOS One's style requirements.

-> Yes. I revised and followed the PLOS One style according to your instructions.

2. Animal experiments (methods of sacrifice, alleviating suffering)

-> Yes, I precisely revised the Materials and Methods section with the following statement: “On the eighth day following the experimental protocol, sacrifice was performed by euthanasia under isoflurane anesthesia.”

3. Amending Competing Interests Statement.

-> Yes. I added “The explicitly states were as below: Employment; KK, SY, TN, TM, and MH were employed by the company. Ownership of stocks; none. Consultancy; none. Patents; granted, pending or applying 20 (applied with Nippon Chemiphar Co. Ltd.). Research grants; Nippon Chemiphar Co. Ltd. Products in development; none. Marketed products; none.

Potassium citrate/sodium citrate (PCSC) was kindly provided by Nippon Chemiphar Co., Ltd. (Tokyo, Japan).

This does not alter our adherence to PLOS ONE policies on sharing data and materials.” in the “Conflict of Interest Statement” section.

4. Role of Funders Statement.

-> Yes. I added the following sentences to the Funding Sources section: “The employees: KK designed animal experiments. SM analyzed the data and prepared the manuscript. TN, TM, and MH contributed to data collection.”

5. Ethics statement location.

-> Yes. As describing the ethical statements including the ID “Animal IRB: AN20190618-04” in the “Material and Methods” section, I deleted it from all other sections.

6. Supporting Information captions.

-> Yes. I revised the contents according to PLOS One guidelines carefully.

7. Reviewer citation recommendations.

-> Yes, I understood.

Responses to Comments of Reviewer #1:

1. Abstract

Introduction under abstract lacks clarity and sentences are disjointed.

-> Thank you. To clarify the study purpose in the Abstract, we revised the sentence "Citrate supplementation alleviates muscle fatigue and prevents renal dysfunction" to: "Citrate supplementation is well known to alleviate muscle fatigue. Furthermore, our previous clinical study revealed that citrate supplementation prevents renal oxidative stress and dysfunction. We hypothesize that an interaction between muscle and kidney tissues underlies the effects of citrate supplementation."

2. Method Authors administered 2000 mg/kg body weight of PCSC by oral gavage to PCSC group daily for one week. What is the rationale for choosing 2000mg/kg/bw?

-> Yes. The oral dose of 2000 mg/kg body weight of potassium citrate/sodium citrate (PCSC) was chosen based on previous studies using normal rodents [Acta Urol Jpn. 1986, 32;1341-1347, Int J Mol Sci. 2025, 26:3329. doi:10.3390/ijms26073329]. We revised the relevant phrase in the Materials and Methods section to: "The PCSC group received 2000 mg/kg body weight of PCSC by oral gavage twice a day for one week. This dose was determined according to previous reports."

3. Why did the authors choose 8 days for the drug/agent administration?

-> Thank you. According to a preliminary study conducted by our collaborating research company, PCSC significantly increased urinary pH within one day, and the increase in urinary pH plateaued after one week. Thus, 8 days of PCSC administration was deemed sufficient for this study [Uric Acid Research 4; 56-64, 1980, DOI: https://doi.org/10.14867/gnam1977.4.1_56]. This preliminary study is in Japanese and has been attached to the "other" section of the PLOS One online submission site.

Additionally, another study demonstrating that PCSC attenuates paclitaxel-induced allodynia showed that the effectiveness could appear within one week [Int J Mol Sci. 2025, 26:3329. doi:10.3390/ijms26073329]. This paper is cited as Reference 6 in the Materials and Methods section.

4. Any reason why authors used only one dose level rather than 3-5 to examine dose-dependency as a proof of cause and effect?

-> Thank you. As explained above, our collaborating research company had preliminarily conducted a dose-dependency study of PCSC on urine pH, blood pH, and base excess in normal rats [Uric Acid Research 1980,4;56-64. doi: https://doi.org/10.14867/gnam1977.4.1_56, and Alkalinize effect of CG-120 on blood and urine in normal and experimental acidotic rats. (J-GLOBAL ID:200902045838196522)]. We therefore applied the chosen dose of 2000 mg/kg body weight of PCSC by oral gavage daily for one week in the current study. Since these preliminary studies were published only in Japanese, they have been attached to the "other" section of the PLOS One online submission site.

5. Statistics Authors calculated muscle growth with the formula: MG= (weight of vastus lateralis muscle)/[(body weight on day 8)- (body weight on day 0)], kindly provide reference. Formula for determining the renal function should be provided.

-> Thank you. We agree with this comment and have revised the formula for assessing muscle gains by PCSC intervention. “Kidney weights and vastus lateralis muscle weights (MW) are calculated by the formula: Organ weight (g/gBW) = [weight of organ]/[body weight on day 8]” This change was incorporated into the " Statistics" section of Materials and Methods.

Additionally, we have changed the statistical analysis from the Wilcoxon test to the Student's t-test. Since the study used homogeneous rats, the t-test, which assumes an underlying normal distribution and is generally more powerful when assumptions are met, is more appropriate than the non-parametric Wilcoxon test. The phrase "Significant differences were defined as p<0.05 by Student's t-test" was revised in the "Statistic" section of Materials and Methods. All data of tables and Figures were revised to the statistic results by t-test.

We also revised the “Effects of PCSC on muscle mass and renal function” section of Results and Discussion with a yellow highlight: "There were no significant differences in the kidney weights between the groups. However, the MW at day 8 was significantly higher in the PCSC group than in the control (mean ± SD: 1.39±0.11 g/100gBW vs. 1.49±0.07 g/100gBW, p=0.0472) (Fig 1 and Table 1)." And p values by t-test analyzed on renal function were corrected “p=0.0027” on control and “p<0.0001” on PCSC group.

Legends of Fig 1 and Fig 2 was corrected accordingly: “The mean diamond indicates mean and 95% confidence intervals”.

6. Results and discussion

Kindly provide brief explanation for box plot. Is the middle line mean or median, the error bars, what quartile if it applies?

-> Thank you. We revised the box plots. Middle lines were indicated the mean values of total data, but the middle lines were unnecessary and eliminated. “The mean diamond indicates mean and 95% confidence interval.” was added in the Legends of Fig 1 and 2.

7. Reference appropriate sections eg., A significant increase in malate levels was observed in the kidneys. Cytoplasmic citrate can be converted to malate by malate enzymes 1 and 3, utilizing NADH as a cofactor. The increased malate/aspartate ratio suggests activation of the malate-aspartate shuttle in renal mitochondria.

-> Thank you for this insightful comment. To make our interpretation clearer, we deleted the initial sentences and revised the "Metabolome analysis of the kidney affected by PCSC" section of Results and Discussion as follows: "Significant increases in the malate level and the Mal/Asp ratio were observed in the kidneys. In the kidney, cytoplasmic citrate can also be converted to malate by these enzymes. The increased malate/aspartate ratio suggests that PCSC could activate the malate-aspartate shuttle in renal mitochondria. This mechanism aligns with findings in a renal ischemia-reperfusion model where the inhibition of malate dehydrogenase-2 attenuated oxidative stress and protected tubular cells by shifting malate metabolism from the mitochondria to the cytosol via the malate shuttle [J Biochem Mol Toxicol. 2024, 38:e23854. doi:10.1002/jbt.23854]. Furthermore, a study in normal rats on high salt intake suggested that the malate-aspartate shuttle enhances energy supply by efficiently delivering cytosolic malate into the mitochondrial matrix, a change associated with suppressed TCA cycle and increased glycolytic enzyme transcription [bioRxiv [Preprint]. 2023, Jan 31:2023.01.18.524636. doi:10.1101/2023.01.18.524636]. The malate-aspartate shuttle is known to contribute to mitochondrial ATP production in the kidney."

Responses to Comments of Reviewer #2:

1. Abstract

• Please ensure that all abbreviations are spelled out in full the first time they appear in the abstract for clarity. This helps readers unfamiliar with the terms to better understand your work.

-> Yes, all abbreviations are spelled out in full the first time. “NADPH/NADP+ ratio” was revised as “Nicotinamide Adenine Dinucleotide Phosphate-hydrogen/Nicotinamide Adenine Dinucleotide Phosphate+ ratio” in the Abstract.

• The abstract/background section appears limited in scope and depth. I recommend expanding it to provide a clearer context for the study, including key motivations, and the specific research gap being addressed. This will help readers better understand the significance and novelty of your work from the outset.

-> Thank you. To provide a clearer context and purpose for the study, we revised the Abstract as detailed: “Citrate supplementation is well known to alleviate muscle fatigue. Furthermore, our previous clinical study revealed that citrate supplementation prevents renal oxidative stress and dysfunction. We hypothesize that an interaction between muscle and kidney tissues underlies the effects of citrate supplementation.

3. Introduction

• The introduction is currently too brief and does not sufficiently establish the physiological or mechanistic link between renal dysfunction and muscle fatigue. To improve the manuscript, I recommend expanding the introduction to include:

• (1) the role of kidney function in metabolic homeostasis,

• (2) how impaired renal function can lead to systemic metabolic disturbances

• (3) the downstream effects of these disturbances on skeletal muscle metabolism and fatigue. (decreased creatinine and NADPH/NADP⁺ ratio may reflect altered energy metabolism and oxidative stress in muscle tissue, while changes in urea and amino acid metabolism (e.g., N,N-dimethylglycine, sarcosine) in kidney tissue could signal impaired nitrogen handling and mitochondrial dysfunction, both of which are known to contribute to fatigue).

-> Thank you. I explained renal dysfunction and metabolic acidosis as “Exercise generates waste products in the body, which are eliminated through urine. Renal dysfunction leads to the accumulation of uremic toxins and metabolic acidosis” in the Introduction. To introduce the muscle atrophy and renal dysfunction, the sentences of “On the other hand, sarcopenia is prevalent in CKD patients and is considered to related with decreased mitochondrial function. [Curr Opin Nephrol Hypertens. 2021, 30:369-376. doi:10.1097/MNH.0000000000000700.]”

The interaction of kidney and muscle was discussed more as described later. The discussion of “The reduction of the muscle NADPH/NADP+ ratio, sarcosine, and creatinine levels by PCSC suggests increased oxidative stress due to exercise promotion. This finding was not inconsistent with the observation that bata-agonists increase sarcosine and creatinine, thereby preventing muscle atrophy in diabetic animal models [Pharmaceutics. 2023, 15:2101. doi:10.3390/pharmaceutics15082101]. The PCSC induced enhancement of muscle metabolism, simultaneously it increased renal urea concentration and activated the betaine-homocysteine metabolic pathway in the kidneys and possibly the liver. Thereby renal function could be improved, and the elimination of the toxins could be promoted. Thus, PCSC is hypothesized to contribute to the improvement of renal function through a collaboration between the muscle and the liver.” was added in the “Summary of metabolomic analysis with significant pathways” section of Results and Discussion.

4. Material and Methods

Animals and experimental protocol

• reference for 2000 mg/kg body weight of PCSC.

-> Yes. I revised the phrase as “The PCSC group received 2000 mg/kg body weight of PCSC by oral gavage twice a day for one week. The dose was decided according to the previous reports” in the Materials and Methods, and referred the papers of Acta Urol Jpn. 32(9);1341-1347 (1986) and Int J Mol Sci. 2025 Apr 3;26(7):3329. doi:10.3390/ijms26073329.

Extraction of metabolites and preparations for metabolome analyses

• The section on “Extraction of metabolites and preparations for metabolome analyses” requires additional methodological detail to ensure reproducibility and transparency.

-> The metabolome analysis was outsourced to Human Metabolome Technologies Inc. (Tsuruoka, Japan) and paid. Therefore, detailed protocols for the metabolome assay are unavailable. The description was revised as “Metabolome analyses were conducted using Human Metabolome Technologies Inc. (Tsuruoka, Japan). Capillary electrophoresistime-of-�ight mass spectrometry (CE–TOFMS) was performed as described previously [J Biol Chem. 2010, 285;39160-70. doi:10.1074/jbc.M110.167304].” in the “Extraction of metabolites and preparations for metabolome analyses” of Materials and Methods. The acknowledgement was written there.

・The Method section should be expanded to include detailed information on animal handling and authentication.

-> Yes. The sentences of “On day 8 following the experimental protocol, sacrifice was performed by euthanasia under isoflurane anesthesia (Mylan Japan Inc., Tokyo, Japan).” was added in the “Animals and experimental protocol” section of Materials and Methods.

5. Results

• I noticed that the pathology figure, which provides valuable visual evidence supporting the study’s findings, is currently included only in the supplementary materials. I strongly recommend moving this figure into the main manuscript. Specifically, it would be appropriate to reference and describe it within both the Methodology and Results sections as separate paragraphs.

-> Yes. The histopathology sections of Materials and Methods, and Results and Discussion were added as below.

The “Histological examination” section and the sentences of “After the sacrifice, the right kidney and the right vastus lateralis muscle from the control and PCSC groups were fixed in 10% neutral buffered formalin. The tissues were then paraffin-embedded, sectioned, and stained with hematoxylin and eosin for microscopic examination.” were added in Materials and Methods.

The “Histological analysis of kidney and muscle” section and the sentences of “No abnormal histological findings were observed in the vastus lateralis muscle or kidneys of either group (Fig 3). It was considered that 2000 mg/kg body weight of PCSCs was safe for muscle and kidney.” in Results and Discussion.

6. Discussion

• Please enrich the Discussion section by adding references to previous studies on octreotide and deferoxamine to better support and contextualize your findings. Including relevant literature will help demonstrate how your results align with or differ from prior work, highlighting the novelty and significance of your study.

-> Thank you for the comment. Regrettably it could be not found the research reports associated with octreotide and deferoxamine regarding this study.

The sentences about kidney metabolism were added as “The significant increases in malate level and Mal/Asp ratio were observed in the kidneys. Pancreatic islets express the enzymes’ genes that cytoplasmic citrate converted to malate; ATP-citrate lyase, malate dehydrogenase

---

## [Editor Report · Decision Letter 1]

2 Dec 2025

Citrate supplement facilitated muscle growth and renal maturation.

PONE-D-25-36940R1

Dear Dr. Abe,

We’re pleased to inform you that your manuscript has been judged scientifically suitable for publication and will be formally accepted for publication once it meets all outstanding technical requirements.

Kind regards,

Sudarshan Kasireddy, Ph.D.

Academic Editor

PLOS ONE

Additional Editor Comments (optional):

The authors have thoroughly addressed the comments raised by the previous reviewers, demonstrating careful attention to detail and a commitment to improving the manuscript. The article, in its current form, meets the standards of quality and scientific rigor expected for publication.
---

## [Editor Report · Acceptance letter]

PONE-D-25-36940R1

PLOS One

Dear Dr. Abe,

I'm pleased to inform you that your manuscript has been deemed suitable for publication in PLOS One. Congratulations! Your manuscript is now being handed over to our production team.

Kind regards,

on behalf of

Dr. Sudarshan Kasireddy

Academic Editor

PLOS One